# First Detection of *Meloidogyne luci* (Nematoda: Meloidogynidae) Parasitizing Potato in the Azores, Portugal

**DOI:** 10.3390/plants10010099

**Published:** 2021-01-06

**Authors:** Leidy Rusinque, Filomena Nóbrega, Laura Cordeiro, Clara Serra, Maria L. Inácio

**Affiliations:** 1Instituto Nacional de Investigação Agrária e Veterinária (INIAV, I.P.), 2780-159 Oeiras, Portugal; leidy.rusinque@iniav.pt (L.R.); filomena.nobrega@iniav.pt (F.N.); 2Direção Regional da Agricultura, Azores, Vinha Brava, 9700-240 Angra do Heroísmo, Portugal; laura.MT.cordeiro@azores.gov.pt; 3Direção-Geral de Alimentação e Veterinária, DGAV, 1349-017 Lisboa, Portugal; cserra@dgav.pt; 4GREEN-IT Bioresources for Sustainability, ITQB NOVA, Av. da República, 2780-157 Oeiras, Portugal

**Keywords:** identification, EST phenotype, root-knot nematodes, mtDNA

## Abstract

Potato is the third most important crop in the world after rice and wheat, with a great social and economic importance in Portugal as it is grown throughout the country, including the archipelagos of Madeira and the Azores. The tropical root-knot nematode (RKN) *Meloidogyne luci* is a polyphagous species with many of its host plants having economic importance and the ability to survive in temperate regions, which pose a risk to agricultural production. In 2019, *M. luci* was detected from soil samples collected from the council of Santo António in Pico Island (Azores). Bioassays were carried out to obtain females, egg masses, and second-stage juveniles to characterize this isolate morphologically, biochemically, and molecularly. The observed morphological features and morphometrics showed high similarity and consistency with previous descriptions. Concerning the biochemical characterization, the esterase (EST) phenotype displayed a pattern with three bands similar to the one previously described for *M. luci* and distinct from *M. ethiopica*. Regarding the molecular analysis, an 1800 bp region of the mitochondrial DNA between cytochrome oxidase subunit II (COII) and 16S rRNA genes was analyzed and the phylogenetic tree revealed that the isolate grouped with *M. luci* isolates (99.17%). This is the first report of *M. luci* parasitizing potato in the Azores islands, contributing additional information on the distribution of this plant-parasitic nematode.

## 1. Introduction

Potato, *Solanum tuberosum*, is the third most important crop in the world after rice and wheat, with more than 156 countries producing it, and hundreds of millions of people depending on it for survival. According to Food and Agriculture Organization of the United Nations FAO estimates, in 2018, over 368 million metric tons of potatoes were produced worldwide, a substantial increase from 333.6 million metric tons in 2010. China is the biggest producer of potatoes worldwide, with an estimated production of 91 million metric tons, while Europe accounts for 106 million metric tons [1]. In Portugal, this crop has great social and economic importance, since it is grown throughout the country, including the archipelagos of Madeira and the Azores. On average, 430,000 metric tons of potato are produced, with the most representative production areas being Bragança, Chaves, Aveiro, Viseu, Oeste Region, and Montijo [2].

Plant-parasitic nematodes are a hampering factor in potato production and quality. Many species have been reported to be associated with potato, among which are the potato cyst nematodes *Globodera* sp., the root-knot nematodes (RKN) *Meloidogyne* sp., the lesion nematodes *Pratylenchus* sp., the potato-rot nematode *Ditylenchus destructor*, and the false root-knot nematode *Nacobbus aberrans* [3]. 

RKN are one of the oldest known parasitic nematodes of plants and considered serious pests of economically important crops [4,5]. The genus comprises more than 90 species [6] and many have been reported in Portugal: *Meloidogyne arenaria* (Neal, 1889) Chitwood, 1949; *Meloidogyne chitwoodi* Golden et al., 1980; *Meloidogyne enterolobii* Yang and Eisenback, 1983; *Meloidogyne hapla* Chitwood, 1949; *Meloidogyne hispanica* Hirschmann, 1986; *Meloidogyne incognita* (Kofoid and White, 1919) Chitwood, 1949; *Meloidogyne javanica* (Trub, 1885) Chitwood, 1949); *Meloidogyne luci* Carneiro et al., 2014; *Meloidogyne lusitanica* Abrantes and Santos, 1991; and *Meloidogyne naasi* Franklin, 1965 [7,8,9,10,11].

*Meloidogyne luci* was first described in 2014 on different plant species in Brazil, Chile, and Iran [12]. Due to its morphological resemblance and similar esterase (EST) phenotype to *M. ethiopica*, several populations of *M. ethiopica* in Europe were reclassified and identified as *M. luci* using biochemical and molecular analyses. In Portugal, it was detected in 2013 in a potato field near Coimbra [9] and was recently found parasitizing tomato, *Solanum lycopersicum,* the ornamental plant *Cordyline australis*, and the weed *Oxalis corniculata* [13]. Since *M. luci* is a polyphagous species with many of its host plants being of economic importance, it poses a risk to agricultural production, especially for potato. Furthermore, its detection in Europe shows that it has the potential to enter the region and survive under temperate conditions [14]. For those reasons, in 2017 *M. luci* was added to the European Plant Protection Organization (EPPO) alert list and in 2019 a national survey was implemented aiming to avoid dispersion. 

The aim of the present study was morphological, morphometric, biochemical and molecularly characterize the isolate of RKN *M. luci* found in the Azores islands.

## 2. Results and Discussion

Morphological characterization from the recovered second-stage juveniles (J2), males and females of the isolate of *M. luci* was performed, as were morphometric studies on J2 (Table 1).

### 2.1. Morphological and Morphometric Characterization

The J2 were vermiform, slender, and clearly annulated. The head region was slightly set apart from the body. The stylet was delicate, narrow, and sharply pointed; the knobs were small and oval shaped. The excretory pore was distinct and the hemizonid was anteriorly adjacent to the excretory pore. The tail was conoid with a rounded tip and the hyaline terminus was distinctive (Figure 1a–d).

Females were elongated, ovoid, or pear-shaped, with a prominent neck (Figure 1d). The head was slightly set apart from the body. The stylet was robust, with knobs well developed. The stylet cone was wider near the shaft and the shaft was wider near the junction with knobs (Figure 1e). The perineal pattern was oval to squarish, with the dorsal arch high to low and rounded. The striae were smooth and wavy, widely separated, and continuous. The lateral lines were weakly demarcated and the perivulval region was free from striae (Figure 1f). The patterns were found to be highly variable. 

The males were vermiform, bluntly rounded posteriorly, and with an anterior end narrowing. The body cuticle was annulated, with large annuli. The head region was not set off from the body. The stylet was robust, and the cone was larger than the shaft and increased in width near the junction with the shaft. The knobs were rounded and small, merging gradually into the shaft. The tail was short and the spicules were curved.

Morphological features are valuable tools for RKN identification due to their low cost and ease of learning the skills, with accuracy depending on the number of characteristics to be evaluated and the number of specimens. The species identification of *Meloidogyne* based on these characters is nevertheless a challenge because morphological differences between RKN species are in most cases indistinctive and measurements of individual specimens in general overlap. 

The morphology and morphometrics were compared to the description of *M. luci* made by [9] and the results were consistent (Table 1). However, due to the intraspecific variability, its identification became difficult; for instance, characteristics such as the morphology of the perineal pattern were highly variable and could be found in more than one species. Therefore, morphological and biometrical diagnostic characteristics need to be supported by other studies, such as biochemical and molecular.

### 2.2. Biochemical Characterization

The EST phenotype from young egg-laying females exhibited three bands (relative mobility, Rm: 1, 1.10, 1.20), corresponding to the *M. luci* L3 phenotype [11]. *Meloidogyne ethiopica* also presented an EST phenotype of three bands (Rm: 0.93, 1.13, 1.24). The three EST bands observed in *M. javanica* (Rm: 1, 1.17, 1.26) were used as a reference to determine the relative position of *M. luci* and *M. ethiopica* bands (Figure 2).

Many studies have shown the usefulness of the nonspecific EST phenotype as the quicker, more reliable, and more stable method to identify *Meloidogyne* spp. [15,16]. The EST phenotype found in the isolate from the Azores was similar to the Portuguese isolate found parasitizing potato in Continental Portugal in 2013. The first band was located at the same level of the band of the reference *M. luci* and *M. javanica*. Since *M. ethiopica* was included for comparison, it could be clearly seen that this first band was well above. Therefore, in spite of the similarity between *M. luci* and *M. ethiopica,* the patterns have clear differences and can be consider reliable in the identification of these two species.

### 2.3. Molecular Characterization

The PCR amplification of mtDNA COII/16S rRNA yielded a single fragment of 1800 bp. The nucleotide sequence obtained in this study was deposited into the GenBank database (NCBI) under the accession number MW160418. A BLAST search of the nucleotide sequence showed a similarity of 99.17% with the sequences of *M. luci* available in the database.

The molecular phylogenetic analysis is presented in Figure 3. The phylogram revealed one clade, supported by a bootstrap value of 93%, that included all isolates of *M. luci* from other countries, the isolate from the Azores, and the isolates of *M. ethiopica*. The isolates of *M. javanica, M. incognita*, and *M. arenaria* formed separate major clades with bootstrap values of 81, 97, and 99%, respectively.

According to [17], the region of mtDNA COII/16S rRNA is useful in the identification of the closely related species *M. luci* and *M. ethiopica*. In this study, that region allowed us to identify the isolate from the Azores as *M. luci*. However, due to the closeness between the species, the molecular markers needed to be used in combination with biochemical analysis.

In general, the eradication of nematodes is very difficult, and it is even more so when it comes to the RKN species, especially *M. luci*. Its ability to adapt to temperate conditions and a wide variety of hosts make its management a challenge. Additionally, several nematicides have recently been strictly regulated or banned in the EU, due to the adverse impacts on the environment and human health, reducing the alternatives for control. Therefore, to define sustainable management strategies, an accurate diagnosis and knowledge of the species is required, with the combination of biochemical and molecular analysis being the best approach for RKN species identification. Furthermore, not only does the identification have constrains, but the detection does as well, as occurred in this study. Plants may either not present any symptoms, or they can often be misdiagnosed, as symptoms may appear similar to other factors. 

Finally, due to the threats and problems presented above, to evaluate the distribution and potential impact of this nematode, a national survey was implemented after 2019 in Continental Portugal and the Azores.

To our knowledge, this is the first report of *M. luci* in the islands of the Azores, Portugal, adding valuable information to the current location of this organism in the EPPO zone.

## 3. Materials and Methods 

### 3.1. Nematode Isolates

During the 2019 National Survey in the Azores islands, soil samples were collected from the council of Santo António on Pico Island. Each consisted of 5 to 8 cores sampled at roughly equal intervals. Six composite soil samples were placed in polyethylene bags and brought for analysis. A 400 mL subsample was taken from each composite sample and the nematodes were extracted using sieving and decanting together with centrifugal technique according to protocol PM 7/119 (1) [18]. The suspension was observed under a stereomicroscope (Nikon SMZ1500, Tokyo, Japan) and suspect specimens of *Meloidogyne* were observed using a bright-field light microscope (Olympus BX-51, Hamburg, Germany) for confirmation.

For positive detections of *Meloidogyne* it was necessary to perform bioassays in order to obtain material (females, egg masses, and males) for identification. Bioassays were carried out by planting tomato plants cv. Oxheart in the remaining soil from the analyzed sample and maintained in a quarantine greenhouse for two months. Females and egg masses were handpicked from the infected tomato roots.

### 3.2. Morphological and Morphometric Characterization 

Nematodes were placed in a drop of water on a glass slide and gently heat killed for morphological and morphometric characterization using a bright-field light microscope (Olympus BX-51, Hamburg, Germany) and photographed with a digital camera (Leica MC190 HD, Wetzlar, Germany). The measurements were taken using the Leica LAS Live. Perineal patterns of adult females were cut from live specimens in 45% lactic acid and mounted in glycerine.

### 3.3. Biochemical Characterization 

Young egg-laying females were handpicked from infected tomato roots and transferred to micro-hematocrit capillary tubes with 5 µL of extraction buffer (20% sucrose *v*/*v* and 1% Triton X-100 *v*/*v*). The females were macerated with a pestle, frozen, and stored at −20 °C until use. Proteins were separated by polyacrylamide gel electrophoresis (PAGE) on thin-slab 7% separating polyacrylamide gels, in a Mini-Protean II (BioRad Laboratories, Hercules, CA, USA) according to [19]. The gels were stained for EST activity with the substrate α-naphthyl acetate. Protein extracts from young egg-laying females of *M. ethiopica* and *M. luci* were included in each gel for comparison and a protein extract of an isolate of *M. javanica* was used as a reference.

### 3.4. Molecular Characterization

The mtDNA COII/16S rRNA region was selected for molecular characterization of the *M. luci* isolate from the Azores islands. The total DNA was extracted from the egg masses using the DNeasy Blood & Tissue kit (Qiagen, Hilden, Germany) following the manufacturer’s instructions. The mtDNA COII/16S rRNA region was amplified using the primers C2F3 (5′-GGTCAATGTTCAGAAATTTGTGG-3′) and 1108 5′-TACCT TTGACCAATCACGCT-3′ [20]. PCR reactions were performed in a 50 μL final volume mixture containing 25 µL Supreme NZYTaq II Green Master Mix, 10 µL of isolated DNA, and 0.2 µM of each primer in a Biometra TGradient thermocycler (Biometra, Göttingen, Germany). Thermal cycling conditions were as described by [17]. PCR products were resolved by electrophoresis at 5 V.cm^−1^ in agarose gel (1.5%) containing 0.5 µg/mL ethidium bromide and 0.5x Tris-borate-EDTA (TBE) running buffer. Amplifications were visualized using the VersaDoc Imaging System (BioRad Laboratories, Hercules, CA, USA). PCR products were purified using the DNA clean and concentrator kit (Zymo Research Corp, Irvine, CA, USA), according to the manufacturer’s instructions. Amplicons were sequenced in both directions at STABVida Sequencing Laboratory (Lisbon, Portugal) on a DNA analyzer ABI PRISM 3730xl (Applied Biosystems). The newly obtained sequence was manually checked, edited, and assembled. The sequence was compared to those of *M. luci* and other relevant sequences of *Meloidogyne* spp. available in the GenBank database using the BLAST homology search. The multiple alignment of the retrieved sequences was performed using ClustalW multiple alignment in BioEdit (Appendix A).

Phylogenetic analyses were conducted using MEGA X v10.1 [21] and the maximum likelihood (ML) method based on the Hasegawa–Kishino–Yano model. The robustness of the ML tree was inferred using 1000 bootstrap replicates. 

## Figures and Tables

**Figure 1 plants-10-00099-f001:**
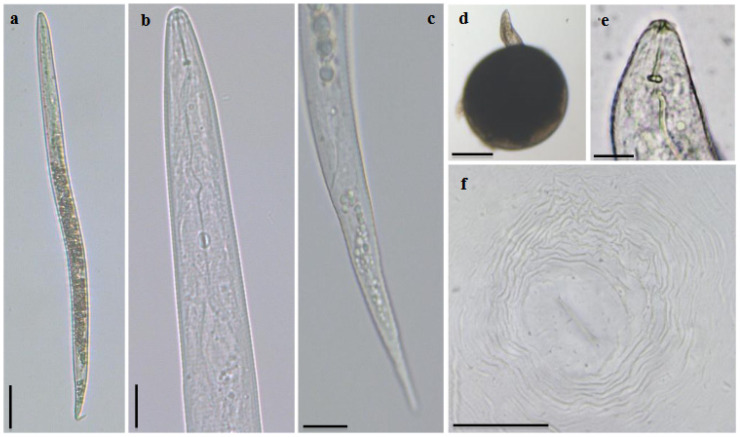
*Meloidogyne luci* light microscope observations. Second-stage juvenile: (**a**) whole specimen; (**b**) anterior region, (**c**) tail region. Female: (**d**) egg-laying female, whole specimen; (**e**) anterior end; (**f**) perineal pattern (bar = 20 µm).

**Figure 2 plants-10-00099-f002:**
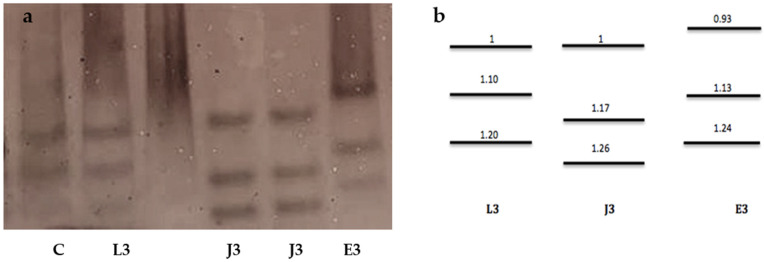
(**a**) Phenotypes of protein homogenates from one egg-laying female of the *Meloidogyne* species: C—Positive control *M. luci*; 1: L3—*M. luci* esterase (Azores), J3—*M. javanica*, and E3—*M. ethiopica*, and (**b**) relative mobility L3—*M. luci*, J3—*M. javanica*, and E3—*M. ethiopica*.

**Figure 3 plants-10-00099-f003:**
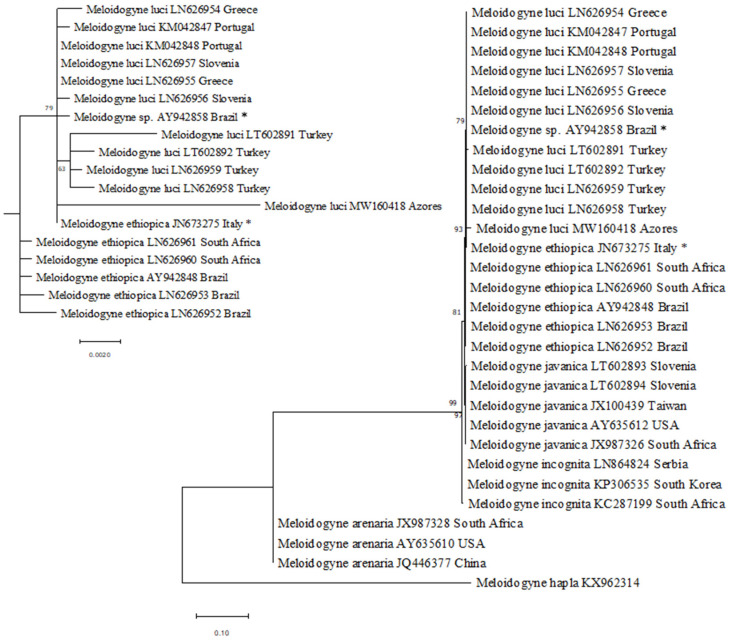
Phylogenetic relationships of *Meloidogyne luci* isolate collected from the Azores, Portugal, and *M. luci* isolates from other geographical regions, including other species of the *Meloidogyne* group, based on the sequence alignment of the mtDNA region between COII and 16S genes. The dendrogram was inferred by using the maximum likelihood method and the Hasegawa–Kishino–Yano model with 1000 bootstrap replication. Bootstrap values are indicated at the nodes. The analysis involved 30 nucleotide sequences and there was a total of 1596 positions in the final dataset. Evolutionary analyses were conducted in MEGA X. * Recently reclassified as *M. luci*, according to [16].

**Table 1 plants-10-00099-t001:** Morphometric comparison of second-stage juveniles (J2) of *Meloidogyne luci* from the Azores, Portugal, with the original description (Carneiro et al., 2014). All measurements are in µm and in the format mean ± standard deviation (range).

Character/Ratio	*M. luci* J2 (Azores) (*n* = 10)	Carneiro et al., 2014 (*n* = 30)
Length	404.99 ± 23 (376.3–446)	383 ± 85 (300–470)
Stylet length	14.05 ± 1 (11.4–15.6)	12.5 ± 0.2 (12.0–13.5)
Dorsal oesophageal gland	3.04 ± 0.41 (2.2–3.4)	2.9 ± 0.5 (2.3–3.3)
Tail length	46.64± 5.92 (39.7–54.1)	44 ± 4.5 (40.0–48.5)
Hyaline terminus	11.23 ± 1.97 (9.02–14.06)	11.7 ± 3.0 (9–15)
Max. body width	15.93 ± 2.26 (14.6–20.9)	16 ± 1.5 (13–20)
a *	25.76 ± 2.87 (22.2–31.9)	25.6 ± 10.5 (15.0–36.1)
c **	8.77 ± 0.86 (7.5–9.6)	8.7 ± 2.6 (6.2–11.5)

* length/max. body width; ** length/tail length.

## Data Availability

The data presented in this study are available in Appendix A: Alignment of *M. luci* isolate from the Azores islands and available sequences on GenBank.

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
