# Peer review of "First Detection of Meloidogyne luci (Nematoda: Meloidogynidae) Parasitizing Potato in the Azores, Portugal"

_plants, 2021, doi:10.3390/plants10010099_

Round 1

Reviewer 1 Report

I enjoyed reading this well-written paper on Meloidogyne luci parasitizing potato plants in the Azores. My impression is that the manuscript needs some slight polishing of the language.

Line 56: Although the abbreviation “EST phenotype” has already been explained in the Abstract, please, repeat your explanation in the Introduction.

Line 62: has the potential

Line 64: a national survey. In Portugal?

Lines 117, 123, 128: Replace “The esterase” or “Esterase” with EST.

Line 160: delete “in order” as redundant.

Author Response

Line 56: Although the abbreviation “EST phenotype” has already been explained in the Abstract, please, repeat your explanation in the Introduction.

We have explained the abbreviation as it was the first time mentioned in the text, as per your suggestion.

Line 62: has the potential

We have changed it accordingly.

Line 64: a national survey. In Portugal?

Yes, a survey in Portugal to detect the presence of the Meloidogyne species included in the A2 alert list has been in place since 2018. This survey is meant to detect M. chitwoodi and M. fallax, and also M. luci. However, we have been asked by our National Plant Protection Organization to provide species identification of any Meloidogyne detected whenever possible.

Line 117, 123,128: Replace “The esterase” or “Esterase” with EST.

We have replaced the full word for the abbreviation EST.

Line 160: delete “in order” is redundant.

Thank you for your suggestion. It has been removed the phrase “in order”.

Reviewer 2 Report

General comments:

The article "First detection of Meloidogyne luci (Nematoda: Meloidogynidae) parasitising potato in the Azores, Portugal" is the first report of this plant-parasitic nematode in the Portuguese islands of the Azores. Although M. luci has already been reported in Portugal, this information is nonetheless useful to the current location of this organism in the EPPO zone, as reported by the authors.

The manuscript is well developed, even if some details are missing especially in the materials and methods and the work would have been more complete if the females had also been measured, but since it is not the description of a new species and having the molecular analysis confirmed the species, the work, in my opinion, can be accepted as it is, without further measurement.

Minor comments:

Keywords

root-knot nematodes (lowercase initial letter for knot)

Introduction

- Pag.2, lines 38 and 40: Please, add references.

- Pag.2, line 40: I suggest changing “the crop…” with “This crop…”

- Pag.2, line 45: I suggest deleting (PCN) since this abbreviation is not mentioned further in the text

- Pag.2, line 56: Please explain EST (It is the first time mentioned in the text)

- Pag.2, line 58: There is one space more between the point and in Portugal.

Results and discussion

- Pag. 3, line 76: The authors wrote in the text that the females are pear-shaped but insert only one figure of a globose female. Maybe they could write as in the original description “elongated, ovoid or pear-shape”. Please, check other female specimens and specify better.

- Pag. 4, line 104-108: I suggest deleting this part. In this work, it is not difficult to identify the species because morphological differences between RKN species are indistinctive (although this is generally the case) but because only a few measurements have been taken.

Materials and methods

- Pag. 7, line 179: PM 7/119 reports different techniques for extracting nematodes from the soil. Please, specify which technique was used in this work.

Author Response

Pag.2, lines 38 and 40: Please, add references

Reference [1] was in the wrong place as it corresponds to the paragraph comprised between lines 38 to 40. It has been corrected accordingly.

Pag.2, line 40: I suggest changing “the crop…” with “This crop…”

It has been changed as per your suggestion.

Pag.2, line 45: I suggest deleting (PCN) since this abbreviation is not mentioned further in the text

It has ben deleted as suggested.

Pag. 3, line 76: The authors wrote in the text that the females are pear-shaped but insert only one figure of a globose female. Maybe they could write as in the original description “elongated, ovoid or pear-shape”. Please, check other female specimens and specify better

You are completely right. We have only included one picture of a female, which is just an example of how the females look like. From the specimens we observed it could be concluded that they may be elongated, ovoid or pear shape, so we have altered as suggested.

Pag. 4, line 104-108: I suggest deleting this part. In this work, it is not difficult to identify the species because morphological differences between RKN species are indistinctive (although this is generally the case) but because only a few measurements have been taken. Although only a few measurements were taken, they were enough to state that the identification of the species through morphology is not easy. So, the paragraph just wants to state that the morphology is useful but not reliable. That’s the reason why we consider it should not be removed.

Pag. 7, line 179: PM 7/119 reports different techniques for extracting nematodes from the soil. Please, specify which technique was used in this work.

Nematode extraction was made using the sieving and decanting together with centrifugal technique. This has been specified in the text as per your request.

Reviewer 3 Report

The study presents the new data on potato parasitic nematode Meloidogyne luci from the area of Azores (Portugal). It is a pest of many plants of economic importance. I am of an opinion that the article fits into scope of Plants and could be published after some major corrections.

Comments:

Keywords: root-Knot – should be root-knot.

Lines 64, 161, 163: whether the authors know the report “

https://gd.eppo.int/reporting/article-6849 (First report of Meloidogyne luci in the Azores (Portugal)”. Please add an appropriate comment to the manuscript (Introduction ?).

Line 50: genus - should be in normal font.

Line 54 etc: [7,8,9,10]- should be [7-10].

Comment to scientific names. A high number of errors, entry not compliant with International Code of Nomenclature for Plants.  Every time when a scientific name is used for the first time in the paper, authors should use proper nomenclature, such as full name, author and date when species was published and described, for example Meloidogyne arenaria (Neal, 1889) Chitwood, 1949.

Common name can be also used. An abbreviation such as M. arenaria or common name should be used further in the paper.

Not always the author are in the parentheses ! Mistakes in names, there must be commas before the names e,g. Chitwoodi  - should be Chitwood, M. arenaria (Neal 1889) – should be M. arenaria (Neal, 1889). Please check all species.

Scientific names (genus, species) must be italicized (line 276).

Line 109: .. made by [9] - this article only describes the morphology, there is no measurement data here.

Line 110: Please add “adults”: The morphology and morphometrics of adult females and males (or adult nematodes) …

Table 1

Please correct the title – should be “Morphometric comparison…. of Carneiro et al. [number] ….” or of [number] ….”.

First column: what does the “L” , “a” and “c” signify?, please explain/change: Length, ratio … .

Line 5, column 3: should be … (40.0-48.5), hyphen.

Line 7: diameter or width ?

Stylet Length or Stylet length F; Dorsal Oesophageal gland or Dorsal Oesophageal Gland; etc. Please unify.

Please change title in the second and third columns: “Second stage juveniles M. luci (n=10)” – should be “Present n=10” and “As per Carneiro et al., 2014” - should be “Carneiro et al. [number] or “[number]”.

Figure 1

Photos were taken using a bright-field light microskope or contrast phase ?. It looks like phase contrast.

g- perianal: should be “perineal”.

Please correct references list, a high number of errors; see Instructions for authors:

  1. Camacho, M.J.; Nóbrega, F.; Lima, A.; Mota, M.; Inácio, M.L. Morphological and molecular identification 243 of the potato cyst nematodes Globodera rostochiensis and G. pallida in Portuguese potato fields. Nematology 2017, 00, 1-7, doi

Should be: “…………………… Nematology 2017, 8, 883–889…”

  1. Dong, K. R. A.; R. A. Fortnum, B. A.; Lewis, S. A. Development of PCR primers to identify species of root knot nematodes: Meloidogyne arenaria, M. hapla, M. incognta and M. javanica. Nematropica 2001, 31, 273 -282.

Should be: Dong, K.; Dean, R.A.; Fortnum, B.A.; Lewis, S.A. Development of PCR primers to identify species of root knot nematodes: Meloidogyne arenaria, M. hapla, M. incognita and M. javanica. Nematropica 2001, 31, 271–280.

  1. Abrantes, I. M. de O.; Santos, M. C. V. dos; Conceição, I. L. P. M. da; Santos, M. S. N. de A.; Vovlas, N. 257 Root-knot and other plant-parasitic nematodes associated with fig trees in Portugal. Nema. Mediterranea 2008, 36:131–6.

Should be: Abrantes, I. M. de O.; Santos, M. C. V. dos; Conceição, I. L. P. M. da; Santos, M. S. N. de A.; Vovlas, N. 257 Root-knot and other plant-parasitic nematodes associated with fig trees in Portugal. Nema. Mediterranea 2008, 36, 131–136.

etc, etc…

Please change the dash/hyphen length for pages.

Please correct abbreviations, dots, etc., e.g. Plant Pathol - Plant Pathol.; J. Nematol - J. Nematol.; Molecular Biology and Evolution – Mol. Biol. Evol.

Author Response

Lines 64, 161, 163: whether the authors know the report “ https://gd.eppo.int/reporting/article-6849 (First report of Meloidogyne luci in the Azores (Portugal)”. Please add an appropriate comment to the manuscript (Introduction ?).

As it is well known, the INIAV works closely with the Portuguese NPPO. As it has been mentioned in the report, INIAV was responsible for the detection and identification of this species and the data used to produce that report is the one we want to publish in this manuscript. That is why we believe it is not relevant to include it.

A high number of errors, entry not compliant with International Code of Nomenclature for Plants.  Every time when a scientific name is used for the first time in the paper, authors should use proper nomenclature, such as full name, author and date when species was published and described, for example Meloidogyne arenaria (Neal, 1889) Chitwood, 1949. Common name can be also used. An abbreviation such as M. arenaria or common name should be used further in the paper. Not always the author are in the parentheses. Mistakes in names, there must be commas before the names e,g. Chitwoodi  - should be Chitwood, M. arenaria (Neal 1889) – should be M. arenaria (Neal, 1889). Please check all species.

We have made the correspondent alterations to standardise the scientific names along the text. The scientific names from line 51 to 55 have been written in full, for instance, Meloidogyne arenaria (Name, year) as suggested.

Line 109: .. made by [9] - this article only describes the morphology, there is no measurement data here

Table 1 shows the comparison between the measurements we made to 10 juveniles of Meloidogyne and the measurements made to 30 juveniles in the initial description of the species by “Carneiro, R. M. D. G.; Correa, V. R.; Almeida, M. R. A.; Gomes, A. C. M. M.; Deimi, A. M.; Castagnone-Sereno, P.; Karssen, G. Meloidogyne luci n. sp. (Nematoda: Meloidogynidae), a root-knot nematode parasitising different crops in Brazil, Chile and Iran. Nematology 2014, 16,289-301, doi. org/10.1163/15685411-00002765.”

Line 110: Please add “adults”: The morphology and morphometrics of adult females and males (or adult nematodes) …

We performed the morphological and morphometric analyses not only in adults but also in juveniles of Meloidogyne. Table 1 shows the biometrics of second-stage juveniles, so, it would be misleading and unlawful to say that we mainly used adults.

Please correct the title – should be “Morphometric comparison…. of Carneiro et al. [number] ….” or of [number] ….”.

The comparison we made was between the second-stage juveniles found in the Azores Island and the second-stage juveniles studied and described by Carneiro et al. It is not a comparison between specimens found by Carneiro et al., so, we believe the title of table 1 explains clearly and concise the data presented in the table.

First column: what does the “L” , “a” and “c” signify?, please explain/change: Length, ratio …

We have added an explanation of the ratios under the table. L=Length has been written in full and a- length/max. body width; c- length/tail length and b= length/tail length

Stylet Length or Stylet length F; Dorsal Oesophageal gland or Dorsal Oesophageal Gland; etc. Please unify

We have uniformed the text as suggested. It now reads Stylet length, Dorsal oesophageal gland and Max. body width.

Please change title in the second and third columns: “Second stage juveniles M. luci (n=10)” – should be “Present n=10” and “As per Carneiro et al., 2014” - should be “Carneiro et al. [number] or “[number]”.

We believe altering the title as suggested would lead to confusion, making difficult to understand what we want to present. We want the readers to know that 10 second-stage juvenile specimens of the population found in the Azores were measured and compared to second-stage juveniles from the description made by Carneiro et al.

Photos were taken using a bright-field light microskope or contrast phase ? It looks like phase contrast.

Photos were taken using a bright-field microscope as it has been mentioned in the text and not with contrast phase.

Please correct references list, a high number of errors; see Instructions for authors

Pages numbers have checked and the references have been corrected following the instructions of the journal.

Please correct abbreviations, dots, etc., e.g. Plant Pathol - Plant Pathol.; J. Nematol - J. Nematol.; Molecular Biology and Evolution – Mol. Biol. Evol.

The abbreviations of the journals have been corrected.

Round 2

Reviewer 3 Report

Introduction:

Scientific names must not be written arbitrarily.

This is regulated i.a. by the International Code of Zoological Nomenclature and Code of Nomenclature for Plants. Please check and change.

Meloidogyne arenaria (Neal, 1889; Chitwood, 1949)

should be:  Meloidogyne arenaria (Neal, 1889) Chitwood, 1949. Only "Neal, 1889" in the parentheses,

or (acceptable notation, only first author/authors) Meloidogyne arenaria (Neal, 1889),

Meloidogyne chitwoodi (Golden et al., 1980)

should be:  Meloidogyne chitwoodi Golden et al., 1980: without parenthesis, please check "Chitwoodi" or "Chitwood" ?,

Meloidogyne hapla (Chitwoodi, 1949)

should be:  Meloidogyne hapla Chitwoodi, 1949: without parenthesis, please check "Chitwoodi" or "Chitwood" ?,

Meloidogyne hispanica (Hirschman, 1986)

should be:  Meloidogyne hispanica  Hirschmann, 1986: without parenthesis, please add “n”,

Meloidogyne incognita (Kofoid and White, 1919; Chitwood, 1949)

should be:  Meloidogyne incognita (Kofoid and White, 1919) Chitwood, 1949: see comment Meloidogyne arenaria,

Meloidogyne javanica (Trub, 1885; Chitwood, 1949)

should be:  Meloidogyne javanica (Trub, 1885) Chitwood, 1949: see comment Meloidogyne arenaria,

Meloidogyne lusitanica (Abrantes and Santos, 1991)

should be:  Meloidogyne lusitanica Abrantes and Santos, 1991: without parenthesis,

Meloidogyne luci (Carneiro et al., 2014)

should be:  Meloidogyne luci Carneiro et al., 2014: without parenthesis,

Meloidogyne enterolobii (Yang et al., 1983)

should be:  Meloidogyne enterolobii Yang and Eisenback, 1983: without parenthesis and wrong author/authors.

Author Response

Dear Reviewer,

Thank you for the careful and thorough reading of our manuscript “First detection of Meloidogyne luci (Nematoda: Meloidogynidae) parasitising potato in the Azores, Portugal”

We appreciate your comments and constructive suggestions, which help us to improve the quality of this manuscript. We have tried to reply to your comments and we are now convinced that our MS meets the requirements to be published in this Journal.

As we were asked, all revised items are marked along the text. Also, the answers to the reviewer comments are written below. 

Thank you once again for your advice and review.

The authors

Response to reviewer #1 comments

Scientific names must not be written arbitrarily. This is regulated i.a. by the International Code of Zoological Nomenclature and Code of Nomenclature for Plants. Please check and change.

Meloidogyne arenaria (Neal, 1889; Chitwood, 1949) - should be:  Meloidogyne arenaria (Neal, 1889) Chitwood, 1949. Only "Neal, 1889" in the parentheses, or (acceptable notation, only first author/authors) Meloidogyne arenaria (Neal, 1889).

Meloidogyne chitwoodi (Golden et al., 1980) - should be:  Meloidogyne chitwoodi Golden et al., 1980: without parenthesis, please check "Chitwoodi" or "Chitwood" ?,

Meloidogyne hapla (Chitwoodi, 1949) - should be:  Meloidogyne hapla Chitwoodi, 1949: without parenthesis, please check "Chitwoodi" or "Chitwood" ?,

Meloidogyne hispanica (Hirschman, 1986) - should be:  Meloidogyne hispanica  Hirschmann, 1986: without parenthesis, please add “n”,

Meloidogyne incognita (Kofoid and White, 1919; Chitwood, 1949) - should be:  Meloidogyne incognita (Kofoid and White, 1919) Chitwood, 1949: see comment Meloidogyne arenaria,

Meloidogyne javanica (Trub, 1885; Chitwood, 1949) - should be:  Meloidogyne javanica (Trub, 1885) Chitwood, 1949: see comment Meloidogyne arenaria,

Meloidogyne lusitanica (Abrantes and Santos, 1991) - should be:  Meloidogyne lusitanica Abrantes and Santos, 1991: without parenthesis,

Meloidogyne luci (Carneiro et al., 2014) - should be:  Meloidogyne luci Carneiro et al., 2014: without parenthesis,

Meloidogyne enterolobii (Yang et al., 1983) - should be:  Meloidogyne enterolobii Yang and Eisenback, 1983: without parenthesis and wrong author/authors.

The scientific names have been corrected in the manuscript as per your suggestion and we really thank for this information.

Dear Reviewer,

Thank you for the careful and thorough reading of our manuscript “First detection of Meloidogyne luci (Nematoda: Meloidogynidae) parasitising potato in the Azores, Portugal”

We appreciate your comments and constructive suggestions, which help us to improve the quality of this manuscript. We have tried to reply to your comments and we are now convinced that our MS meets the requirements to be published in this Journal.

As we were asked, all revised items are marked along the text. Also, the answers to the reviewer comments are written below. 

Thank you once again for your advice and review.

The authors

Response to reviewer #1 comments

Scientific names must not be written arbitrarily. This is regulated i.a. by the International Code of Zoological Nomenclature and Code of Nomenclature for Plants. Please check and change.

Meloidogyne arenaria (Neal, 1889; Chitwood, 1949) - should be:  Meloidogyne arenaria (Neal, 1889) Chitwood, 1949. Only "Neal, 1889" in the parentheses, or (acceptable notation, only first author/authors) Meloidogyne arenaria (Neal, 1889).

Meloidogyne chitwoodi (Golden et al., 1980) - should be:  Meloidogyne chitwoodi Golden et al., 1980: without parenthesis, please check "Chitwoodi" or "Chitwood" ?,

Meloidogyne hapla (Chitwoodi, 1949) - should be:  Meloidogyne hapla Chitwoodi, 1949: without parenthesis, please check "Chitwoodi" or "Chitwood" ?,

Meloidogyne hispanica (Hirschman, 1986) - should be:  Meloidogyne hispanica  Hirschmann, 1986: without parenthesis, please add “n”,

Meloidogyne incognita (Kofoid and White, 1919; Chitwood, 1949) - should be:  Meloidogyne incognita (Kofoid and White, 1919) Chitwood, 1949: see comment Meloidogyne arenaria,

Meloidogyne javanica (Trub, 1885; Chitwood, 1949) - should be:  Meloidogyne javanica (Trub, 1885) Chitwood, 1949: see comment Meloidogyne arenaria,

Meloidogyne lusitanica (Abrantes and Santos, 1991) - should be:  Meloidogyne lusitanica Abrantes and Santos, 1991: without parenthesis,

Meloidogyne luci (Carneiro et al., 2014) - should be:  Meloidogyne luci Carneiro et al., 2014: without parenthesis,

Meloidogyne enterolobii (Yang et al., 1983) - should be:  Meloidogyne enterolobii Yang and Eisenback, 1983: without parenthesis and wrong author/authors.

The scientific names have been corrected in the manuscript as per your suggestion and we really thank for this information.
